

# Assessing blue orchard bee (*Osmia lignaria*) propagation and pollination services in the presence of honey bees (*Apis mellifera*) in Utah tart cherries

Natalie K. Boyle and Theresa L. Pitts-Singer

USDA ARS Pollinating Insects Research Unit, Logan, UT, United States of America

## ABSTRACT

*Osmia lignaria* is a commercially available, native solitary bee species recognized for its propensity to forage upon and pollinate tree fruit crops such as apple, almond and cherry. This study evaluated the implementation of *O. lignaria* co-pollination with honey bees in central Utah commercial tart cherry orchards during 2017 and 2018 bloom. Three paired 1.2 ha sites were selected for evaluation of cherry fruit set and yield with and without managed *O. lignaria* releases alongside the standard honey bee hive stocking rate of 2.5 hives/ha. *Osmia lignaria* supplementation did not measurably increase cherry fruit set, fruit per limb cross-sectional area or fruit weight. The lack of differences in yield is likely a consequence of local saturation of pollinator services supplied by managed honey bees throughout experimental orchards, such that no additive benefit of managed *O. lignaria* releases were measurable. An increase in managed *O. lignaria* populations was achieved in 2017 but not 2018, possibly due to unknown changes to orchard management or environmental factors. While flying *O. lignaria* in Utah tart cherries may support sustainable in-field bee propagation, their subsequent impacts on tart cherry yield were not detected when paired with standard stocking densities of honey bees.

**Submitted** 6 April 2019
**Accepted** 6 August 2019
**Published** 5 September 2019

**Corresponding author**
Natalie K. Boyle,
natalie.boyle@ars.usda.gov

**Academic editor**
Ilaria Negri

**Additional Information and Declarations can be found on page 11**

**DOI 10.7717/peerj.7639**

# INTRODUCTION

Fruit and nut tree crops comprise an important and critical segment of US agricultural production, which relies heavily, and often exclusively, upon insect pollination to achieve profitable yields. Cherries are the eighth most valuable tree fruit crop, valued at nearly $950 million in 2017 (*Perez & Minor, 2018*). With a 36% increase in tart cherry yield projected for 2018 from the previous year (*Perez & Minor, 2018*), it is increasingly important to consider the role and scope of insect pollination in the industry. As weather patterns during early spring blooms become less predictable, heavy rains and late frosts are expected to have measurable impacts on fruit yield (*Cannell & Smith, 1986*; *Houston et al., 2018*), in part because such inclement conditions do not support efficient honey bee foraging behavior. More than ever, growers may be able to benefit from the supporting wild and alternative

managed insect pollinators to achieve the pollination requirements for profitable cherry production.

While tart cherries do not require insect pollination to set fruit, previous work has demonstrated up to five-fold increases in fruit set with insect-mediated cross-pollination (*Shoemaker, 1928*; *Free, 1993*). Most commercially available varieties are self-compatible, such that pollen transferred from an anther to the stigma of the same blossom can yield fruit. In the absence of insect pollination, tart cherry blossoms are reliant on wind and other abiotic factors for pollen transfer. Regardless, it is generally acknowledged that tart cherry orchards frequently benefit from added pollination services provided by insects, and as such, rented honey bee colonies are usually placed in tart cherry orchards during bloom.

The use of honey bees as commercially managed pollinators in monoculture is a standard choice that makes sense for growers, considering their ease of management, ease of transport and their well-studied biology. However, given recent concerns over honey bee health and availability due to various known and potentially unknown risk factors (*vanEngelsdorp et al., 2009*), there is now growing interest in supporting alternative wild and managed pollinators, whose contributions to commercial agriculture have historically been overlooked (*Garibaldi et al., 2013*; *Isaacs et al., 2017*).

The blue orchard bee, *Osmia lignaria* Say (Hymenoptera: Megachilidae), is among the most widely studied native solitary bee species in the United States. Unlike honey bees, *O. lignaria* have only one generation per year and are free-flying adults for just four to six weeks annually (*Bosch & Kemp, 2001*). *Osmia lignaria* are cavity-nesters and will reside gregariously in artificially-supplied hollow cavities such as reeds, corrugated wood blocks or cardboard tunnels (*Bosch & Kemp, 2001*). Females exhibit a foraging preference for rosaceous crops such as apples, almonds and cherries (*Torchio, 1979*; *Torchio, 1985*; *Bosch & Kemp, 1999*; *Bosch & Kemp, 2001*; *Sheffield, 2014*), collecting pollen and nectar to construct provision masses for their offspring. Each egg is laid on top of its own provision, and the mother completes each cell by sealing it with a mud partition before the next provision mass is created (*Bosch & Kemp, 2001*). Under favorable conditions, one female can provision and lay eggs in over 15 cells in her lifetime (*Bosch & Kemp, 2001*). *Osmia lignaria* are spring-flying bees that forage in cooler temperatures than honey bees (*Bosch & Kemp, 2001*), making them well-suited for early-blooming and/or high elevation crops. The timing of their activity can be easily and actively manipulated through established cold storage incubation practices, such that they can be available to pollinate orchards any time between mid-February and late May. They also collect dry (unwetted) pollen underneath their abdomen (in coarse hairs that are collectively referred to as a 'scopa'), which facilitates pollen deposition and cross-pollination to flowers visited. Their anatomy and behavior on flowers yield more efficient pollination than that of honey bees, who store wetted pollen tightly-packed in pollen baskets, i.e., corbiculae (*Parker et al., 2015*). In fact, 275 female *O. lignaria* can provide pollination services comparable to a strong (>8 frames) honey bee colony (*Bosch & Kemp, 2001*). Finally, *O. lignaria* provide localized pollination services, as they do not typically forage beyond 100 m from their nest site (*Bosch & Kemp, 2001*), as long as local floral resources are abundant. In contrast, honey bees generally forage within

3.2 km of their hive (*Eckert, 1933*) and have been observed foraging at distances upwards of 9.5 km (*Beekman & Ratnieks, 2000*).

Until recently, the biggest challenge for commercial-scale pollination by *O. lignaria* has been their limited supply. Growing public interest in wild and alternative bees over the past decade has led to the development of a competitive *O. lignaria*-based industry, from which the necessary quantities of bees are now available for field-scale trials and uses (*Boyle & Pitts-Singer, 2017*; *Andrikopoulos & Cane, 2018*; *Pitts-Singer et al., 2018*; *Pinilla-Gallego & Isaacs, 2018*). Recent studies have and continue to identify the crops and geographic regions in which managed *O. lignaria* populations best perform. Such examples include in California almond orchards, where co-pollination with a honey bees (at a full stocking density) with *O. lignaria* result in significantly improved fruit set and nut yield versus when either pollinator is used alone (*Brittain et al., 2013*; *Pitts-Singer et al., 2018*). We hypothesized that similar benefits of *O. lignaria* pollination in tart cherry orchards may be obtainable. The current study reports on the potential bee reproductive success and influence of *O. lignaria* co-pollination with honey bees on commercial tart cherry production in central Utah.

## MATERIALS AND METHODS

### Experimental set-up and bee management

This trial was conducted over a contiguous 101 ha swath of Montmorency cherries on Mahelab rootstock in Santaquin, UT during spring 2017 and 2018. Verbal permission to access and use the site was obtained via phone conversations and meeting in person with the property (orchards) owner. All orchards were 10–15 years in age and managed using the same general practices. Three square 1.2 ha plots (110 × 110 m; 'OL+') were selected to receive *O. lignaria* alongside honey bee hives (2.5 hives/ha). To serve as a control, three paired 1.2 ha plots were additionally selected, where only honey bees (2.5 hive/ha; 'OL-') were available for pollination (a randomized complete block design). Experimental plots were separated by at least 100 m to minimize potential spillover effects of *O. lignaria* foraging in controlled OL- plots (Fig. 1). Honey bee hives (2 deeps per hive) were placed along orchard edges on pallets (four hives per pallet) to achieve the desired stocking density; hives were owned and managed directly by contracted beekeepers.

Within the selected OL+ sites, eighteen nest boxes were uniformly distributed throughout each 1.2 ha plot (5 boxes per row in 3 equally dispersed rows; Fig. 1). Nest boxes were made of dark blue corrugated plastic boxes (21.5 × 20 × 25.5 cm), each housing one-hundred 7 × 152 mm cardboard nesting tunnels lined with waxed paper straws. Each nest box was hung from a tree branch (ca. 1.2 m above ground) with a southeast-facing entrance, in accordance with established best management practices (*Bosch & Kemp, 2001*). Prior to their deployment, all nest boxes were treated with a spray-on application of a patented chemical bee attractant (*Pitts-Singer et al., 2016*) to promote bee nesting. No nesting accommodations were made for *O. lignaria* in OL- orchard plots.

*Osmia lignaria* used in this study were acquired from a commercial supplier in Northern Utah (both years); none of the 2017 progeny were allocated to the 2018 study. Bees were

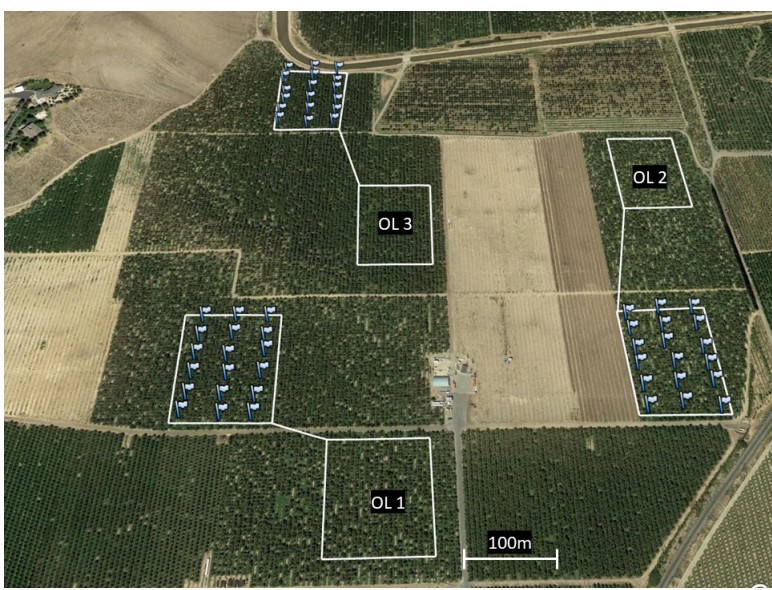

**Figure 1  Aerial map of three paired OL+/OL- locations selected for 2018.** Each flag corresponds to the location of an artificial *O. lignaria* nest box (6 per row across 3 equally dispersed rows). Nest boxes were placed only in OL+sites; empty white squares without flags correspond to OL–sites. Managed bee releases took place at the center of each OL+site.

stocked in OL+ orchard plots at a density of 625 females/ha, per the recommendation of *Bosch & Kemp (2001)*. Bee release sites were located at the center of the three OL+ plots where just-emerged adults and about-to-emerge adults in cocoons were placed in black fiberglass boxes (40 × 25 × 20 cm), situated ∼12 cm above ground. Adult *O. lignaria* exited the release boxes naturally through a small hole at the bottom of each box. For both years, approximately 40% of the bees had emerged from their cocoons following an overnight 'preincubation' at room temperature prior to orchard release. Bloom occurred on 29 April 2017 and 27 April 2018, and our corresponding *O. lignaria* releases took place on 1 May 2017 and 30 April 2018, after the first 15% of cherry blossoms had opened and were receptive to pollination.

Osmia lignaria were left to mate and provision nests in the OL+ treatments for four to five weeks until just before the first scheduled post-bloom fungicide spray (however, *O. lignaria* populations were exposed to a single nutritional spray during late May in 2017 and 2018). No alternative floral resources were added for foraging adults in the orchards. While some wild flowering weeds were present along orchard margins or between rows (e.g., dandelions), heavy chemical control of orchard grounds greatly reduced the availability of non-cherry blossoms. Details of spray timing and tank mixes were not shared with the authors of this study. All nest boxes were returned to the laboratory, where nests were stored at room temperature while offspring completed their development to adulthood. Individual *Osmia lignaria* nests were examined using X-radiography frequently through late summer and early fall until all progeny had become adults, at which time the bees were stripped from their paper tunnels, cleaned of frass, loose pollen and pollen mites,

and moved into cold storage (4 °C) as loose cocoons throughout the late fall and winter months. Progeny recovered in 2017 weere released for various other projects in orchards >250 km from these experimental orchards.

## Bee reproduction

All partial or fully completed *O. lignaria* nests were counted by nest box and X-radiographed (6s exposure at 22 kVp; Faxitron 43804N, Faxitron Bioptics, Tuscon, AZ). From the X-radiographs, the total number of cells in each nest were counted and assessed individually for survival, sex, parasitism and cause/stage of death where applicable (as described in *Boyle & Pitts-Singer, 2017*; *Pitts-Singer et al., 2018*). We also counted 'pollen balls', which is when the provision mass remains uneaten due to the absence of an egg or the failure of that egg to hatch in the cell. Due to differences by year in orchard plot site locations and climate, inter-year statistical comparisons of *O. lignaria* progeny were not made. This decision was made in part due to the 2018 removal and subsequent replanting of tart cherry acreage after the 2017 season. Annual differences in climate were evaluated by qualitatively comparing daily temperature and precipitation during and just following bloom, as provided by a neighboring weather station, located just 4.0 km SSE from our research plots (*US Climate Data, 2019*).

## Fruit production

To assess differences in 2017 and 2018 fruit set and yield between OL+ and OL- treatments, 20 trees from each 1.2 ha plot were randomly selected at bloom and assessed for percent fruit set, fruit weight and fruit per limb cross-sectional area (fruit/LCSA). One limb from each tree was marked and assessed for all three measurements, although the criteria for limb selection varied slightly between years. In 2017, limb selection was confined to branches growing from the trunk of the tree and selected based on approximate length (1–2 m), aspect (SE-facing branch), and height above ground (1.3–2 m). In 2018, limb selection criteria were modified so it would be possible to select a limb of a larger branch. Thus, limbs did not need to stem from the trunk of the tree. Other criteria in 2018 dictated limb length (∼1 m), age (third year growth), aspect (SE-facing branch) and height (1.3–2 m above ground). At the proximal end of each selected limb, the limb circumference was measured ($cm^2$) to allow for fruit/limb cross-sectional area calculations. Because of the change to limb selection metrics between years, cross-year comparisons were not made.

The fruit set was measured using similar to methods employed in *Pitts-Singer et al. (2018)*. All flowers occurring on selected limbs were counted and recorded during bloom. Then, just before harvest (13 July 2017 and 16 July 2018), we returned to the same trees and counted how many cherries had developed on selected limbs to calculate percent fruit set. The total number of cherries on selected limbs were also incorporated into fruit per limb cross-sectional area measurements. For fruit weight, 20 cherries from each randomly selected tree/limb were collected and weighed as pooled samples in the laboratory. Differences in OL+ and OL− fruit set, fruit/LCSA and weight were statistically analyzed separately by year in a randomized complete block design via two-way ANOVA in JMP (*SAS Institute, 2015*; main effects being OL+/OL−, or 'bees', and plot).

**Table 1** **Summary table of 2017 and 2018 *O. lignaria* nesting and progeny outcomes in a Utah tart cherry orchard.** Mean (±SEM) per 1.2 ha plot is reported, followed by the percent progeny outcome in parentheses (where applicable).

| Mean ± SEM (%) | 2017 | 2018 |
|---|---|---|
| Tunnels filled (%) | 38.4 ± 5.7 | 10.0 ± 2.1 |
| Cells recovered | 220.1 ± 40.2 | 62.0 ± 13.6 |
| Cells/Tunnel | 5.1 ± 0.13 | 5.6 ± 0.33 |
| Sex ratio (M:F) | 1.85 | 2.67 |
| Females | 69.5 ± 13.1 (30.5%) | 13.1 ± 3.2 (20.3%) |
| Males | 128.4 ± 23.8 (56.3%) | 35.1 ± 7.8 (54.3%) |
| Immature dead[a] | 6.4 ± 1.3 (2.8%) | 8.3 ± 2.0 (12.8%) |
| Pollen ball[b] | 13.3 ± 2.2 (10.4%) | 8.0 ± 1.7 (12.4%) |
| Parasitized | 2.4 ± 1.1 (>1%) | 0.03 ± 0.01 (>1%) |
| % Female return | 111% | 32% |

**Notes.**
[a] Immature dead includes all cells that failed to develop to become viable adults for unknown reasons. This column pools larval, pupal and adult mortality, as confirmed by X-radiography.
[b] Pollen ball occurs when the provision remains uneaten in the cell, likely due to egg failure.

# RESULTS

## Bee reproduction

The production of bee progeny varied by year. Cell outcome and additional summary data (cells produced, cells/tunnel and sex ratio) are presented in Table 1. In 2017, more viable female progeny had been recovered than were released into the orchard (111% female recovery). In contrast, 2018 female recovery was much lower (32%). The proportion of recovered cells containing viable offspring also differed across years: In 2017, 90% of all cells recovered contained viable progeny, while just 75% of recovered 2018 cells were viable. Some of the 2018 loss in progeny may be attributable to an increase in nests containing uninterrupted provision masses extending throughout the length of the nesting tunnel in the absence of any eggs, mud partitions, or mud plugs at the end of the tunnel (Fig. 2). In 2018, this runaway pollen-collecting behavior was observed in 29% of all nesting tunnels (versus <1% in 2017). There was also a relatively higher rate of progeny in 2018 that died during larval/pupal development, along with a proportionally higher incidence of individual pollen balls. Along with much lower overall recovery of bees in 2018, there were proportionally fewer viable females recovered relative to males (see sex ratios in Table 1).

## Climate data

Weather conditions during cherry bloom favored warmer overall temperatures in 2018 (average daily high/low temperature during bloom: 21.3/7.4 °C) than in 2017 (average daily high/low temperature during bloom: 19.7/5.9 °C). Levels of precipitation during bloom were similar (7.5 cm in 2017 versus 5.3 cm in 2018), although most precipitation received in 2017 came down in a single day as snow (5.1 cm), whereas 2018 precipitation came down gradually as rain on nine different days. Additionally, during 2017 bloom, there were four days in which freezing temperatures were recorded, while no freezes occurred in 2018.

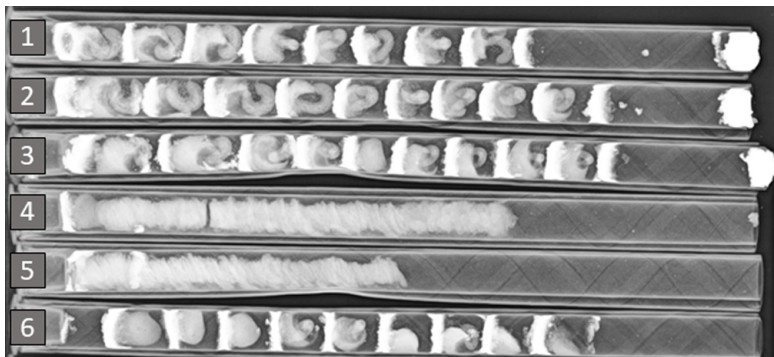

**Figure 2** **Example of an X-radiograph (June 2018) used to assess *O. lignaria* nest contents after recovery from the field.** This image reveals the contents of 6 nesting tunnels. Tunnels 1, 2, 3, and 6 depict developing bees, where provisions, larvae, mud partitions and a mud plug at the terminus (tunnels 1-3 only; right side) of the nest are clearly visible. Tunnels 4 and 5 depict extended, singular provision masses in which no eggs, larvae, mud partitions or plugs are present. This unusual pollen-collecting behavior was observed in 29% of all nesting tunnels used in 2018 (versus <1% in 2017).

**Table 2** **Summary table of 2017 and 2018 tart cherry outcomes when *O. lignaria* were (OL+) and were not (OL−) introduced as co-pollinators with managed honey bee colonies at a rate of 2.5 hives/ha.** Mean (±SEM) per 1.2 ha plot is reported with main effects statistics for each variable measured. Limb cross-sectional area (LCSA) was recorded for the proximal end of selected tree limbs, along the length of which developed fruit were counted to assess Fruit/LCSA.

| | $\bar{X}$ OL− (±SEM) | $\bar{X}$ OL+ (±SEM) | Variable | F | P |
|---|---|---|---|---|---|
| 2017 Fruit Set (%) | 23.9 ± 2.0 | 22.2 ± 2.0 | Bees | 1.0611 | 0.305 |
| | | | Plot | 16.6798 | <0.0001 |
| 2018 Fruit Set (%) | 30.3 ± 1.9 | 29.8 ± 2.0 | Bees | 0.0727 | 0.788 |
| | | | Plot | 2.328 | 0.1019 |
| 2017 Fruit/LCSA (cm$^2$) | 14.2 ± 2.1 | 15.9 ± 1.0 | Bees | 0.3436 | 0.5588 |
| | | | Plot | 3.7403 | 0.0265 |
| 2018 Fruit/LCSA (cm$^2$) | 5.5 ± 0.6 | 6.2 ± 1.0 | Bees | 1.0169 | 0.3153 |
| | | | Plot | 0.7724 | 0.4642 |
| 2017 Fruit Wt (g) | 4.0 ± 0.1 | 4.1 ± 0.1 | Bees | 0.98 | 0.3241 |
| | | | Plot | 5.3988 | 0.0057 |
| 2018 Fruit Wt (g) | 3.6 ± 0.1 | 3.5 ± 0.1 | Bees | 4.73 | 0.0316 |
| | | | Plot | 11.05 | <0.0001 |

## Fruit production

Statistical analyses did not support our hypothesis that *O. lignaria* would influence fruit set or yield in this study (Table 2). Fruit set and yield, by plot (±SEM) and year, are presented in Fig. 3. Overall, the specific location of the paired OL+/OL− experimental plots within the orchard had a significant effect on fruit set and yield, while the implementation of *O. lignaria* did not (Table 2; Fig. 3). In 2018, fruit weight was significantly higher in OL− versus OL+ treatments ($F = 4.73$, $P = 0.0316$); cherry weight was 2.8% heavier when *O. lignaria* were *not* introduced.

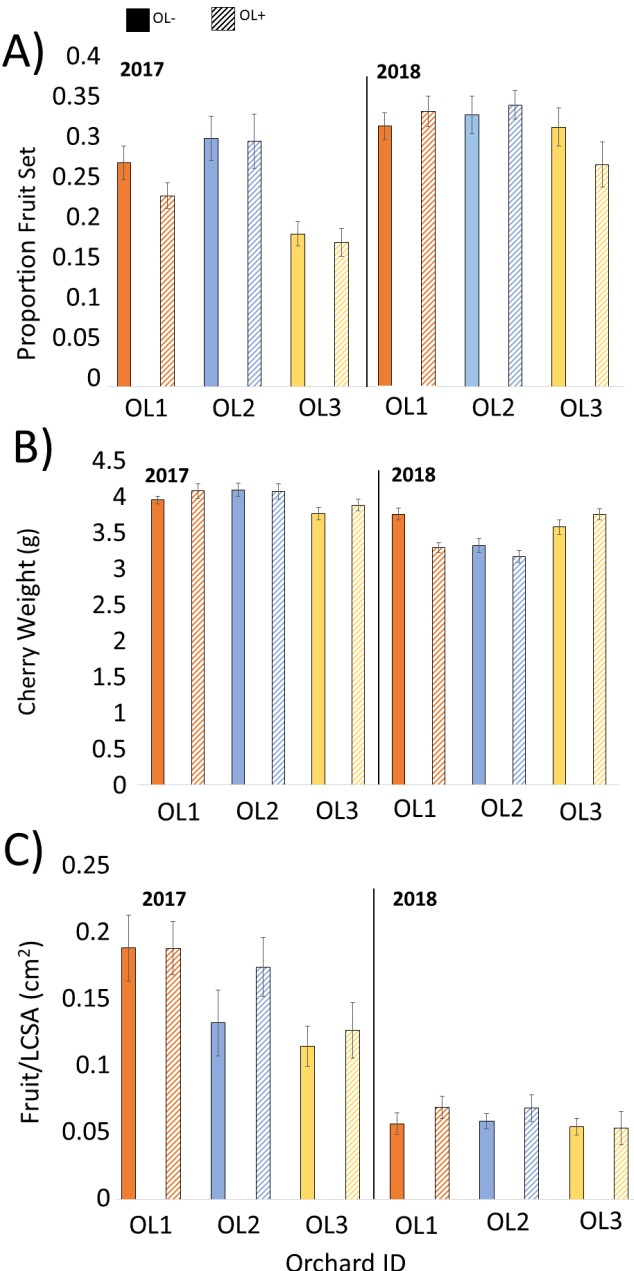

**Figure 3** Mean (±SEM) values for each paired 1.2 ha orchard site in 2017 and 2018 pollination trials comparing honey bee pollination versus honey bee and *O. lignaria* co-pollination in Utah tart cherries. Solid bars signify OL−, or *A. mellifera* only, while hashed bars signify OL+, or *A. mellifera* with *O. lignaria* sites for A) proportion fruit set, (B) individual cherry weight and (C) fruit per limb cross-sectional area (LCSA). Lower fruit/LCSA obtained in 2018 is a consequence of the adoption of new limb selection criteria only.

## DISCUSSION

Our data demonstrated that *O. lignaria* pollination in this scenario had only a very limited, and not reproducible, impact on tart cherry pollination services (significantly heavier cherries were obtained from OL- plots versus OL+ plots in 2018). Previous work in almonds has demonstrated a functional pollination synergy when *O. lignaria* pollination is used in tandem with the traditional stocking density of honey bee hives (5 hives/ha; *Brittain et al., 2013*; *Pitts-Singer et al., 2018*). However, unlike with almonds, tart cherries are self-pollinating, such that insect pollination is not required for trees to bear fruit. Regardless, it is known that insect pollination can significantly increase yields even in self-compatible cropping systems (*Erickson et al., 1978*; *Lansari & Iezzoni, 1990*; *Sabbahi, DeOliveira & Marceau, 2005*), which is why honey bees are frequently rented by tart cherry producers during bloom.

Fruit set measurements in 2017 and 2018 were typical for Montmorency cherries (*Shoemaker, 1928*), suggesting that pollination was not a limiting factor in any of the OL+ or OL- plots evaluated in this study. The lack in consistent statistical differences between OL+ and OL- plots imply that the employed stocking rate of managed honey bee colonies already maximize the potential for high rates of fruit set. *Shoemaker (1928)* reports the fruit setting rate of Montmorency trees to be approximately 6% in the absence of cross-pollination and 25–30% with hand- or insect-pollination. Thus, 25–30% fruit set is the maximum fruit set that can be expected in the present study. Because 2017 and 2018 fruit set ranged from 22–30%, there is no doubt that insect-mediated pollination maximally benefitted tart cherry production. *Ashman et al. (2004)* suggests that after a known pollinator density threshold is reached, the additive effect of increasing pollinators in a given orchard does not improve yields at harvest. This is likely the case in the current study, such that 2.5 honey bee hives/ha is sufficient in meeting pollination demands of tart cherries without the addition of managed *O. lignaria* populations. Further, there is a possibility that foraging *O. lignaria* may have traveled beyond the reported 100 m foraging range, which may interfere with our interpretation of treatment groups. However, as central-place foragers, we find it unlikely that foraging adults would regularly and preferentially access blossoms that fall outside their typical flight range, especially considering the uniformity and high abundance of resources in the local vicinity. From these results, we are not able to gauge the effectiveness of *O. lignaria* as cherry pollinators when no adjustments to honey bee stocking density are made.

No net gain in cherry yield from the release of managed *O. lignaria* populations was observed, and differences between years in fruit/LCSA as presented in Table 1 are merely a consequence of the revision to limb selection criteria in 2018. We found it interesting that the specific location of paired OL+/OL- plots within experimental orchards provided more substantial effects on yield than pollinator inputs, highlighting that even within a relatively uniform geographic area, microclimatic conditions, or slight variations in irrigation or soil profile, may significantly influence fruit yield. For example, Fruit/LCSA in 2017 and 2018 is highest in the northernmost paired plots, although the trend does not carry over for corresponding fruit weight and set metrics.

Regardless of the lack of significant differences detected, alternative pollination may still serve as an appropriate 'pollinator insurance' against unplanned honey bee colony shortages or in conditions where adverse weather may not be conducive to honey bee foraging. Cooler and wetter days are known to have measurable impacts on fruit set potential (*Choi & Anderson, 2001*) along with decreased pollinator activity (*Vincens & Bosch, 2000*), and *O. lignaria* readily forage at cooler morning temperatures than honey bees do (*Bosch & Kemp, 2001*). 2017 was overall a cooler year than 2018, although this did not appear to yield any meaningful differences in the efficacy of fruit set and yield for this study. We propose that there would be value in conducting future studies which examine *O. lignaria* impacts on yield in orchards lacking honey bees, or with fewer honey bee hive stocking densities.

Sustainable *O. lignaria* reproduction was achieved in 2017 but not 2018, and the 2018 failure in bee success is probably tied to the runaway pollen-collecting behavior observed in many of the nesting tunnels that year. While we occasionally observe this behavior by *O. lignaria* in nesting tunnels, the frequency at which we observed this runaway pollen-collecting in 2018 was exceptional and somewhat alarming. The cause of this behavior is unknown but may reflect the quality/condition of the bees that were released, environmental factors, pesticide use, or changes to orchard management. Private *O. lignaria* pollination consultants with contracts in neighboring orchards during 2018 did not observe this behavior (K Clark, 2018, pers. comm.). It is worth noting that the seemingly abnormal behavior of some bees still resulted in active contributions to cross-pollination of cherry blossoms during bloom, despite the limited progeny recovered in 2018.

The ability to achieve sustainable and reliable in-orchard bee return is a known challenge to the management of this bee. Today's supply of *O. lignaria* is met by collecting cocoons from native populations in trap nests placed in wildlands across the Western US (*Tepedino & Nielson, 2017*). This practice escalates environmental and financial costs associated with employing *O. lignaria* as alternative pollinators. Historically, bee return from ochards can vary substantially between years, with typical returns in commercial almond orchards with a full honey bee stocking density ranging 30–40% (*Artz et al., 2013*; *Artz et al., 2014*; *Pitts-Singer et al., 2018*). *Boyle & Pitts-Singer (2017)* observed a 2-fold increase in bee reproduction in a 2016 Utah tart cherry orchard, and in 2017, 1.1× more progeny was recovered at the end of bloom than bees were released. We propose that, despite the low reproduction obtained in 2018, tart cherry orchards may serve as an appropriate avenue for future open field, managed *O. lignaria* propagation efforts, while also providing local pollination services to growers. Future studies are needed to determine if *O. lignaria* can be used as supplements or replacements for honey bees when there is a need or grower decision to employ fewer honey bee hives for crop pollination.

## ACKNOWLEDGEMENTS

We thank lab technicians Ellen Klomps and Craig Huntzinger, along with lab assistants Hannah Jarvis, Penina Meatoga, Jenna Hanson, Alex Foster, and Emily Slingerland for assisting with set-up and installation of field materials, collection of bloom data, and X-ray

diagnoses of *O. lignaria* nests. We also thank Utah State University students Andi Kopit, Morgan Dunn and Alan Anderson for their help in experimental set-up and data collection in the orchards.

### Funding
The authors received no funding for this work.

### Competing Interests
The authors declare there are no competing interests.

### Author Contributions
- Natalie K. Boyle conceived and designed the experiments, performed the experiments, analyzed the data, contributed reagents/materials/analysis tools, prepared figures and/or tables, authored or reviewed drafts of the paper, approved the final draft.
- Theresa L. Pitts-Singer conceived and designed the experiments, performed the experiments, contributed reagents/materials/analysis tools, authored or reviewed drafts of the paper, approved the final draft.

### Field Study Permissions
The following information was supplied relating to field study approvals (i.e., approving body and any reference numbers):

Ray Rowley of Cherry Hill Farms granted verbal authorization for this fieldwork.

### Patent Disclosures
The following patent dependencies were disclosed by the authors:

Bee attractants, filed by TL Pitts-Singer, WP Kemp, D Moreland, SS Peterson, JS Buckner and MM Hagen under U.S. patent number US20140271536A1, were applied to nest boxes to encourage *O. lignaria* nesting in artificial cavities.

### Data Availability
The raw data collected for this trial is available in the Supplemental File.

### Supplemental Information
Supplemental information for this article can be found online at http://dx.doi.org/10.7717/peerj.7639#supplemental-information.

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
