# Peer review of "Assessing blue orchard bee (*Osmia lignaria*) propagation and pollination services in the presence of honey bees (*Apis mellifera*) in Utah tart cherries"

_PeerJ, doi:10.7717/peerj.7639_

## Round 0.1 · original submission · Major Revisions

This study is of great value because it considers the role of wild and alternative bees for tree fruit crop pollination. However, some revisions are needed prior to publication.

First of all, the title of the article does not reflect what actually addressed in the research. Indeed, the pollination service of Osmia lignaria cannot be properly evaluated given the “massive” presence of Apis mellifera foragers. This should be also better highlighted in the discussion, and the conclusion and abstract should be re-written (I agree with the reviewer 1 that a logical option would have been to perform the study in orchards without A. mellifera, but we are all aware that this is very difficult...maybe considering an orchard without hives or with hives far from it?).

I also think that data on A. mellifera colonies should be included. Indeed, the authors supplemented orchards with 625 females of O.lignaria (by the way, how many males?) because, according to Bosch & Kemp (2001), 275 females of O. lignaria can provide pollination services comparable to a full honey bee colony, but authors didn’t provide info on the status of the honey bee families.

In addition, the authors didn’t consider the possible spill-over effect of Osmia bees flying on OL- plots, that can affect control data.
Another interesting point is if in 2018 the authors used the same bees released in 2017 or not: this should be declared and considered in the discussion.

Regarding the weird behaviour displayed by Osmia in 2018, I think that if this was due to the quality of individuals bought and released, the whole work would lose significance...

Given all the considerations above, my decision is major revisions.

Reviewer 1 ·

Basic reporting

In the study, the authors set out to determine the impact that native, commercially managed Blue Orchard Bees (BOBs) have on crop yield in tart cherry production in orchards that were seemingly saturated with non-native commercially managed honey bees. I feel that the manuscript itself is very well-written, and I have no issues with grammar, etc. The structure of the manuscript, use of references, figures and tables are all suitable.

The authors provide context in the introduction as to why this is important in that native pollinators are worth developing and evaluating as managed pollinators of crops, in part because in supports native species, but more importantly, that they may help relieve some of the burden on crop production associated with honey bee health and availability. They also point out that tart cherry is a self-compatible crop that, while not requiring insect pollinators to produce fruit, benefits greatly from insect pollinators, with up to a 5-fold increase in fruit set (though what this represents in total fruit set is not mentioned: What is the maximum fruit set for this crop? What is a good versus poor yield?).

Experimental design

This study was done through the use of paired orchards, some receiving BOBs in addition to honey bees, with control orchards not receiving BOBs, thus being pollinated by honey bees only. Unfortunately, the authors do not really describe what a good yield looks like for cherry production, i.e., were no differences observed with the addition of BOBs because 30% fruit set represents the maximum?

To me, the experimental design does not address the title of this paper, as we do not really see what BOBs are capable of doing on their own in the pollination of this crop in Utah. A logical additional option was to have orchards with BOBs, but without honey bees. This is likely hard to do, as honey bees are virtually ubiquitous, and there are few placed in the USA where one can study crop pollination without honey bees being present (whether brought in, or not). Thus, BOBs are not really evaluated in a way that increases our understanding of their potential in this crop system, or that promotes them as a viable alternative managed pollinator of this crop to honey bees, as it appears that they do not have an impact even though they used the crop floral resources for nest provisions. The take home message for growers may unfortunately be to just use honey bees for tart cherry pollination. As such, I do not feel this study really offers a suitable evaluation of BOBs as tart cherry pollinators.

Validity of the findings

In this study, no large differences in fruit set and weight were found in orchards with and without BOBs, though seemingly the presence of BOBs lowered these in one year. These results suggest that pollination was likely maximized by honey bees alone, and supplemental pollination (i.e., addition of BOBs) above the honey bee stocking rate used here is not necessary, and in fact is likely a waste of money for growers. Interestingly, it seems that tart cherry has a maximum fruit set of 25-30%, though the ideal range for crop production and profit is not mentioned. Presumably this is good level, and the bees provided full pollination, and other internal (i.e., physiological) factors prevent the trees from obtaining a higher yield.

Additional comments

In addition to my comments above:

In the abstract and elsewhere, it is a good idea to indicate that Osmia lignaria is a native, commercial pollinator.
After first mention of Osmia lignaria (or any species), it is okay to use “O. lignaria”, but the standard practice is to not use the abbreviated form at the beginning of a sentence (e.g., lines 19, 60). Thus, use the full form at the beginning of a sentence, and use the shortened form in the middle of a sentence (e.g., line 26). This should be done consistently.
Line 33, eighth
Line 40, remove “additionally”
Line 42, “studies” is plural, but only one study (a 90 year old study in fact) is cited.
Line 73. Is there a reference which supports that an abdominal scopa increases pollination?
Line 132. “incidence of pollen ball” is odd wording, and perhaps just state “We were also able to identify uneaten pollen provisions from the images…” or something similar, or put “pollen ball” in quotes, as you use this code in table 1.
Line 168. 111% female recover versus 32%. Under ideal conditions, what are BOBs capable of in terms of population recovery? Do you feel that the presence of honey bees had a negative impact on BOB recovery? On line 119 you indicate that the BOBs were left in the orchard for five weeks; when were the honey bees removed, or was there continued competition for food even after the crop finished flowering? What did BOBs feed on after cherry?

Reviewer 2 ·

Basic reporting

This manuscript asses the effect of introducing Osmia lignaria in addition to the normal honey bee stock, on yield in cherry orchards. This is an important assessment because, beyond being effective at pollinating flowers, farmers need to know if there is economic benefit (increase in yield) of using this alternative pollinator. The manuscript is written in good English. The literature review is good, although some of the citation could be more recent, and some of the information presented need citations. Figures are good and clear, axes labels of fig 3 could be improved. See my comments on the manuscript. The raw data is available.

Experimental design

The research questions are not explicitly stated in the text, which is something that the authors could include. The aim of this study is important, because if we want farmers to implement alternative pollinators we have to prove that they are adding benefits. I think the authors could do a better job at explain why assessing of O. lignaria in conjunction with honey bees is important and how it fits into the bigger goal of implementing alternative pollinators as a rutting practice in orchards (filling the knowledge gap). It would be good if the authors stated their hypothesis in the introduction.
The methods are clear and detailed enough to replicate. Although some reports find that O. lignaria won’t fly more than 50 m when they have their preferred resources close to their nest, they can fly more than 600 m, as the authors stated in the introduction. OL+ plots were 100 m away from OL- plots, so it is possible that O. ligaria from OL+ plots could have flown to the OL- plot, which will confound the effect of the treatment and the control. This problem could have been address by making observation on the OL- plot during bloom to make sure there were not visits from O. lignaria. I recognize the difficulty in finding an orchard big enough that would allow the proper speciation between plots, or finding multiple growers that would be willing to allow researched to run experiments in their properties, and I agree with the authors in that this is probably a case of pollination saturation due to honey bees, but I think the authors should acknowledge and discuss the possibility of the treatment-mix (see comments on text).

Validity of the findings

The findings of this study are of great value for the alternative pollinators management body of knowledge, as it shows that even when alternative pollinators exist and can be managed, the successful implementation in orchards and crops will require an integrated system than also include honey bees in the equation. This study counts with two years of data, so the results seem to be consistent. The data analysis sounds appropriate, but the authors could be more descriptive on the model used to analyzed the data (see comments on text). The conclusions are clear and based on the results obtained, and the authors propose steps to follow in future projects. Once the authors clarify the research questions/objectives, they could be linked to the conclusions.

Additional comments

Use Adobe reader to see my comments on the manuscript

Annotated reviews are not available for download in order to protect the identity of reviewers who chose to remain anonymous.

---

## Round 0.2 · accepted · Accept

The current version of the manuscript has been significantly improved both in the discussion and conclusion. The title now addresses what actually assessed by authors, i.e. the pollination service and propagation of managed Osmia lignaria in presence of Apis mellifera in tart cherries. The manuscript demonstrates that in orchards saturated by managed honey bees, the introduction of solitary bees does not provide additive benefit on tart cherries yield. Even if we all agree that a proper evaluation of the pollination service of O. lignaria on tart cherries would require orchards without Apis mellifera (and this is quite impossible given that honey bees are ubiquitous and beekeeping is widespread), I personally believe that this work indeed represents an interesting step forward in the knowledge of the fascinating ecological relationships between “managed” species (including co-pollination service) in agroecosystems, and should be accepted for publication.